# TRIBE: TRImodal Brain Encoder
# for whole-brain fMRI response prediction

**Stéphane d'Ascoli**
Meta AI
sdascoli@meta.com

**Jérémy Rapin**
Meta AI
jrapin@meta.com

**Yohann Benchetrit**
Meta AI
ybenchetrit@meta.com

**Hubert Banville**
Meta AI
hubertjb@meta.com

**Jean-Rémi King**
Meta AI
jeanremi@meta.com

## ABSTRACT

Historically, neuroscience has progressed by fragmenting into specialized domains, each focusing on isolated modalities, tasks, or brain regions. While fruitful, this approach hinders the development of a unified model of cognition. In this technical report, we introduce TRIBE, the first deep neural network trained to predict brain responses to stimuli across multiple modalities, cortical areas and individuals. By combining the pretrained representations of text, audio and video foundational models and handling their time-evolving nature with a transformer, our model can precisely model the spatial and temporal fMRI responses to videos, achieving the first place in the Algonauts 2025 brain encoding competition with a significant margin over competitors. Ablations show that while unimodal models can reliably predict their corresponding cortical networks (e.g. visual or auditory networks), they are systematically outperformed by our multimodal model in high-level associative cortices. Currently applied to perception and comprehension, our approach paves the way towards building an integrative model of representations in the human brain. Our code is available at https://github.com/facebookresearch/algonauts-2025.

## 1 INTRODUCTION

**Motivation.** Progress in neuroscience has historically derived from an increasing specialization into cognitive tasks and brain areas. In the domain of vision, for instance, research focused on specialized cortical areas and their associated tasks, such as motion perception in V5 (Shadlen & Newsome, 2001), face recognition in the fusiform gyrus (Kanwisher & Yovel, 2006), or the visual processing of written language in the visual word form area (Dehaene & Cohen, 2011). While this divide-and-conquer approach has undeniably yielded deep insights into the brain's mechanisms of cognition, it has led to a fragmented scientific landscape: How neuronal assemblies together construct and globally broadcast a unified representation of the perceived world remains limited to coarse conceptual models (Mashour et al., 2020).

The fast progress in AI in the domains of language (Brown et al., 2020; Grattafiori et al., 2024), image (Oquab et al., 2023), audio (Baevski et al., 2020; Chung et al., 2021) and video (Tong et al., 2022; Assran et al., 2025) may help resolve this fragmentation challenge. Indeed, the representations learnt by these AI models have been shown to – at least partially – align with those of the brain (Huth et al., 2016; Schrimpf et al., 2018; Caucheteux & King, 2022). Motivated by this unexpected alignment, several teams have built encoding models to predict brain responses to natural stimuli from the activations of neural networks in response to images (Yang et al., 2023; Adeli et al., 2023; Nguyen et al., 2023), speech (Millet et al., 2022) and text (Toneva & Wehbe, 2019). However, these encoding models are currently limited in three critical ways.

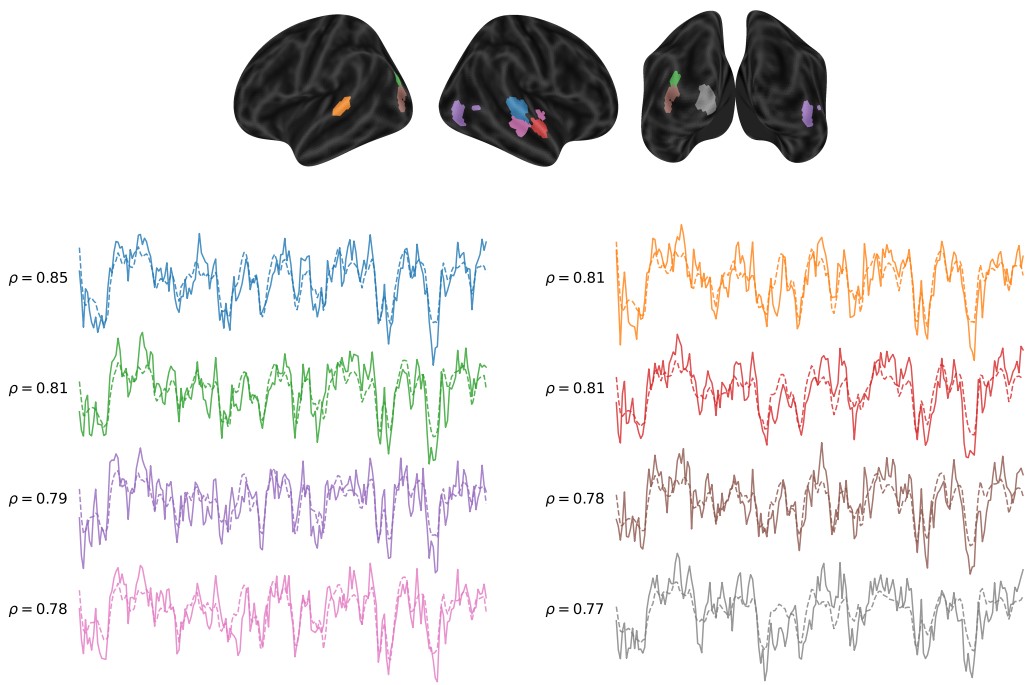

Figure 1: **TRIBE predicts brain responses to videos across diverse regions.** For eight brain parcels color-coded in the brain images, we report the BOLD response of the first participant to the first 5 minutes of a held-out movie in solid lines and our model's predictions in dashed lines, with the Pearson correlation of the two curves reported on the left.

First, *linearity*: existing encoding approaches typically rely on a ridge regression to map the AI model representations onto those of the brain. This assumes that these two sets of representations are linearly equivalent – a likely incorrect assumption (Linsley et al., 2025).

Second, *subject-specificity*: due to large variability in brain responses from one subject to another, existing encoding approaches typically train a separate model for each subject, which prevents them from leveraging the similarities between brains.

Third, *unimodality*: most existing encoding approaches predict brain responses from unimodal stimuli, which makes them incapable of capturing how the brain integrates information from multiple modalities (Hu & Mohsenzadeh, 2025). This is particularly limiting as it has been shown that cross-modal interactions occur not only in specific multisensory areas (Gao et al., 2023; Beauchamp, 2005), but also in primary sensory areas (Driver & Noesselt, 2008; Stein & Stanford, 2008).

**Contribution**   In this work, we introduce TRIBE, a novel deep learning pipeline to predict the fMRI brain responses of participants watching videos from the corresponding images, audio and transcript. This approach addresses the three limitations outlined above: our model learns how to capture the dynamical integration of modalities in an end-to-end manner across the whole brain, and from multiple subjects.

Our model achieves state-of-the-art results, reaching the first place out of 267 teams in the Algonauts 2025 competition on multimodal brain encoding. We observe that the benefit of multimodality is highest in associative cortices, and demonstrate via ablation analyses the importance of the multimodal, multisubject and nonlinear nature of TRIBE.

**Related work**   While there has been recent research on deep learning for multimodal brain decoding (Dahan et al., 2025; Scotti et al., 2024; Xia et al., 2024; Zhou et al., 2024; Kong et al., 2024), there currently exists no equivalent for brain encoding. Some recent works suggest to train recurrent

models to predict brain responses from frozen visual or linguistic features (Güçlü & Van Gerven, 2017; Chehab et al., 2021), or fine-tune existing pretrained models using the brain encoding objective. While these relax the linearity assumption, they are restricted to a single sensory modality.

Conversely, a few recent studies have built encoding models on top of vision-language transformers, demonstrating gains compared to unimodal transformers (Dong & Toneva, 2023; Oota et al., 2022; Doerig et al., 2022; Wang et al., 2022; Tang et al., 2023). However, these works rely solely on linear mappings to model brain responses from the activations of the multimodal transformers. We believe this can be suboptimal for two reasons. First, multimodal transformers are still relatively new: at rare exceptions (Jaegle et al., 2021; Srivastava & Sharma, 2024; Abdin et al., 2024), they often only integrate static images and text (audio and video being significantly more compute-intensive), and tend to lag behind the performance of unimodal transformers. Second, and more fundamentally, the way these models integrate information across modalities may be very different from how the human brain does such multimodal integration. An ideal encoding pipeline should thus *learn* how to best combine different modalities.

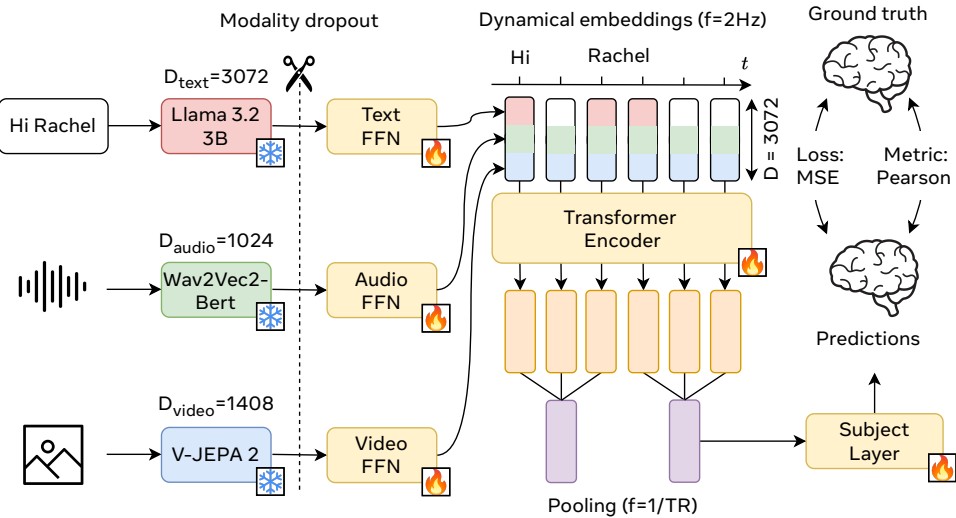

Figure 2: **Visual summary of our method.**

## 2 METHODS

### 2.1 OVERVIEW

**Task**  Our objective is to predict the brain activity of participants watching videos. This is framed as a regression task where the targets are the blood-oxygen-level-dependent (BOLD) signals detected at every repetition time (TR=1.49s) of a 3T fMRI recording device, separated into 1,000 non-overlapping parcels fig. 1. For this, we take as input the video clip being viewed by the participant, as well as the corresponding audio file and transcript. From these, we extract time-series of high-dimensional embeddings from the intermediate layers of start-of-the art generative AI models along three modalities of interest: text, audio and video, which we feed to our deep encoding model, as illustrated in fig. 2.

**Evaluation**  To assess performance, we evaluate for each parcel the Pearson Correlation $\rho$ between predicted and ground truth fMRI responses across all TRs of the evaluation set. We then average these values over all parcels. We also refer to this metric as the "encoding score" of the model. We hold out 10% of the recording sessions for the validation set, ensuring that the same videos are held out for each participant to prevent any form of data leakage.

## 2.2 DATASET

**Data collection**   We train our encoding model on the Courtois NeuroMod dataset (St-Laurent et al., 2023). This dataset consists of six human participants who watched the same naturalistic videos, namely the first six seasons of the popular TV series *Friends* as well as four movies: *The Bourne Supremacy, Hidden Figures, The Wolf of Wall Street* and *Life* (a BBC Nature documentary). This amounts to an unprecedently large recording volume of over 80 hours of fMRI per subject. In the present work, we focus on a subset of four subjects curated for the Algonauts 2025 competition (Gifford et al., 2024).

**Preprocessing**   We use the preprocessing pipeline provided by the Algonauts2025 competition. The whole-brain BOLD fMRI responses are preprocessed using fMRIprep (Esteban et al., 2019) and projected to the Montreal Neurological Institute (MNI152NLin2009cAsym) standard space (Brett, 2002). Functional images are then parcellated by averaging voxel-wise BOLD signals within each of the 1,000 parcels of the Schaefer atlas (Schaefer et al., 2018), yielding a single fMRI time series for each of the parcels. Finally, activations were z-scored per parcel across each session (corresponding to approximately 15 minutes of recording).

## 2.3 MODEL

**Timed text embeddings**   We extract "timed" text embeddings from the timestamped transcripts of the videos. For each word $w$ to embed, we prepend the preceding $k = 1,024$ words in the transcript, which we feed through Llama-3.2-3B (Grattafiori et al., 2024). For each intermediate layer $l$, we extract the token(s) overlapping with the word $w$ and average them to obtain a contextualized word embedding of dimension $D_{\text{text}} = 3072$. We then construct an evenly spaced grid at a frequency $f = 2\,\text{Hz}$, and for each time-bin $\tau$, we sum the embeddings of words which overlap with the bin. This allows to temporally align the text features with the audio and video features.

**Audio embeddings**   To obtain audio embeddings, we extract audio files from the videos, split them into 60-second chunks, then feed these through Wav2Vec-Bert-2.0 (Chung et al., 2021). We then resample the hidden representations of the latter from 50 Hz to $f = 2\,\text{Hz}$. For each intermediate layer $l$, this yields time series of embeddings of dimension $D_{\text{audio}} = 1024$. Note that the resulting embeddings carry bidirectional information about both the past and future of the stimulus window, whereas text and video embeddings only contain information about the past.

**Video embeddings**   For video embeddings, we again construct an evenly spaced grid at a frequency $f = 2\,\text{Hz}$, and for each bin of time, we feed 64 frames spanning the preceding 4 seconds to Video-JEPA 2 gigantic (Assran et al., 2025). For each intermediate layer $l$, we compress the tensor of activations by averaging over all patch tokens, yielding a time series of embeddings of size $D_{\text{video}} = 1408$. Note that this spatial averaging step was necessary to keep the size of the tensor manageable. However, it comes at the cost of discarding positional information, which we expect to deteriorate encoding performance in low-level visual areas which exhibit a retinotopic mapping (Wandell & Winawer, 2011).

**Combining the modalities**   For each of the three modalities $m$, the feature extraction described above leads to a time series of embeddings at $f = 2\,\text{Hz}$, with embeddings of shape $[L_m, D_m]$, where $L_m$ and $D_m$ are the number of layers and dimensionality of the transformer of modality $m$. To compress these embeddings while retaining both low-level and high-level information, for each modality, we split the layers into $L$ groups, then average the tensor per group along the layer dimension, compressing to a shape $[L, D_m]$. As shown in fig. 7, for each modality, the embeddings coming from deeper layers of the corresponding foundational model yield better encoding performance, especially in associative cortices (fig. 8), and the best configuration was found to be $L = 2$ groups ranging from relative depths 0.5 to 0.75, and 0.75 to 1. We then concatenate the layers and feed the resulting vector through a linear layer with a shared output dimension $D = 1024$ followed by layer normalization. Finally, we concatenate the three modalities, leading to a time series of *multimodal* embeddings of shape $3 \times 1024$. This will be the input to our transformer encoder.

**Extracting windows**   From these time series of stimulus embeddings, we extract windows of duration $T = N \times TR$, as well as the corresponding $N$ target BOLD responses. Since the number of input tokens is $N \times TR \times f$, at fixed memory budged, there is a tradeoff between increasing the stimulus sampling frequency $f$ and the window duration $N$. Through a grid search with frequencies ranging from 0.5 to 10 Hz at fixed number of input tokens, we found $f = 2$ Hz and $N = 100$ to yield the best results.

**Transformer encoder**   We add to the stimulus embeddings learnable positional embeddings, then feed the result through a Transformer encoder with 8 layers and 8 attention heads. This enables information to be exchanged between timesteps. At the output of the transformer, we use an adaptive average pooling layer to compress the sequence from length $fT$ to $N$, yielding one embedding per TR.

**Subject layer**   Brain responses to a given stimulus vary from one brain to another. To enable training a single model across all subjects, we use a subject-conditional layer, following (Défossez et al., 2023), which selects a different linear projection for each subject. The latter projects the output of the transformer to the 1,000-dimensional target space. Note that all timesteps in the window are predicted simultaneously, which makes our model particularly compute-efficient at inference.

**Hemodynamic lag**   To avoid border effects due to the hemodynamic lag between the stimuli and the corresponding brain responses, we offset targets by 5 seconds relative to the inputs. Note that since the transformer has access to all timesteps, we do not follow the common practice in linear encoding of convolving the inputs by a temporal response function. Instead, the attention mechanism selects the most relevant timesteps: an analysis presented in fig. 9 shows that the attention weights peak between 5 and 10 seconds relative to the current timestep, which is consistent with the expected hemodynamic response function.

## 2.4   TRAINING DETAILS

**Modality dropout**   One desirable property of a multimodal encoding model is its ability to provide meaningful predictions in the absence of one or several modalities, for example for a silent movie or a podcast. To encourage this behaviour, while at the same time avoiding excessive reliance on one modality, we introduce modality dropout: during training, we randomly mask off each modality by zeroing out the corresponding input tensor with a probability $p$, ensuring that at least one modality is left unmasked.

**Optimization**   We train our model for up to 15 epochs with the AdamW optimizer (Loshchilov & Hutter, 2017) using a batch size of 16. The learning rate is warmed up linearly to $10^{-4}$ over the first 10% of steps, then decayed following a cosine learning rate schedule. We use early stopping based on the validation Pearson score computed on a held-out set. To improve generalization, we use stochastic weight averaging (Izmailov et al., 2018), which involves averaging model weights obtained at the end of each epoch, once the validation metrics are near their plateau.

**Ensembling**   To further improve generalization, we ensemble the predictions of $M = 1000$ models, whose initialization and shuffling seeds are all different. To strengthen ensemble diversity, for each model, we sample a set of hyperparameters uniformly in the grid specified in table 3. For each parcel separately, we compute the encoding score of all models on the validation set, then compute a softmax distribution over models with temperature 0.3, which will determine the weight assigned to each model for this given parcel.

**Implementation details**   We extract stimuli features from pretrained language, audio and video models available on the `HuggingFace` platform (Jain, 2022) and cache them as `Numpy` memmap arrays (Harris et al., 2020) for fast loading during the training of our encoding model. These models respectively contain 3B, 600M and 700M parameters, and feature extraction is completed in 24 hours on 128 V100 GPUs with 32GB of VRAM. The TRIBE model itself contains contains 980M trainable parameters, and the full training loop lasts 24 hours on a single such GPU. We use the transformer

implementation from the `x-transformers` package[1]. We list the licenses of the assets used in this work in section .

| Rank | Mean score | Subject 1 | Subject 2 | Subject 3 | Subject 5 |
|---|---|---|---|---|---|
| 1 (Ours) | **0.2146 ± 0.0312** | 0.2381 | 0.2105 | 0.2377 | 0.1720 |
| 2 | **0.2096 ± 0.0283** | 0.2353 | 0.2046 | 0.2268 | 0.1718 |
| 3 | **0.2094 ± 0.0215** | 0.2233 | 0.2072 | 0.2271 | 0.1798 |
| 4 | **0.2085 ± 0.0267** | 0.2295 | 0.2003 | 0.2300 | 0.1743 |
| 5 | **0.2055 ± 0.0291** | 0.2306 | 0.2010 | 0.2240 | 0.1662 |

Table 1: **Our model achieves first place in the Algonauts 2025 leaderboard.** We report the mean score ± standard deviation for the top five out of 267 teams.

| OOD | Movie | Mean score | Subject 1 | Subject 2 | Subject 3 | Subject 5 |
|---|---|---|---|---|---|---|
| ✗ | Friends Season 7 | **0.3195 ± 0.0289** | 0.3419 | 0.3239 | 0.3346 | 0.2775 |
| ✓ | Pulp Fiction | **0.2604 ± 0.0137** | 0.2765 | 0.2611 | 0.2431 | 0.2610 |
| ✓ | Princess Mononoke | **0.2449 ± 0.0572** | 0.2816 | 0.2507 | 0.2851 | 0.1623 |
| ✓ | Passe-partout | **0.2323 ± 0.0525** | 0.2763 | 0.2587 | 0.2370 | 0.1573 |
| ✓ | World of Tomorrow | **0.1924 ± 0.0323** | 0.2210 | 0.1606 | 0.2196 | 0.1686 |
| ✓ | Planet Earth | **0.1886 ± 0.0380** | 0.1483 | 0.2029 | 0.2331 | 0.1699 |
| ✓ | Charlie Chaplin | **0.1686 ± 0.0551** | 0.2249 | 0.1289 | 0.2080 | 0.1128 |

Table 2: **Our model generalizes to highly out-of-distribution movies.** We report the mean score ± standard deviation for the held-out datasets of the Algonauts 2025 competition.

## 3 RESULTS

### 3.1 ALGONAUTS 2025 COMPETITION RESULTS

Our model achieves the first place out of 267 teams in the Algonauts 2025 competition on multimodal brain encoding. As shown in table 1, we outperform competitors by a substantial margin: the gap between our model and the runner-up is larger than between the runner-up and the fifth.

We display the results across the various held-out datasets in table 2. In the in-distribution setting of the first phase of the competition (*Friends* season 7), our model achieves a mean score of 0.3195. Unsurprisingly, when tested in the out-of-distribution setting of the second phase of the competition, performance is lower, with an average of 0.2146. Remarkably, our model achieves robust scores even in the extreme out-of-distribution setting of cartoons (0.1924 for *World of Tomorrow*) wildlife documentaries (0.1886 for *Planet Earth*), and silent black-and-white movies (0.1686 for *Charlie Chaplin*), where language processing cortices are very poorly encoded (see fig. 10).

### 3.2 WHOLE-BRAIN ACCURACY

How is the prediction accuracy distributed spatially? We report the Pearson scores across the cortical surface in fig. 3. The distribution is highly non-uniform (panel a), and peaks in the temporal and occipital areas of the brain associated with auditory, language and visual processing (panel b). To assess which brain parcels are predicted significantly better than chance, we ran a block-wise permutation test (1000 permutations) across the 164 recording sessions of the held-out validation set. Remarkably, all 1000 parcels are significantly predicted ($q(\text{FDR}) < 10^{-3}$).

---

[1] https://github.com/lucidrains/x-transformers

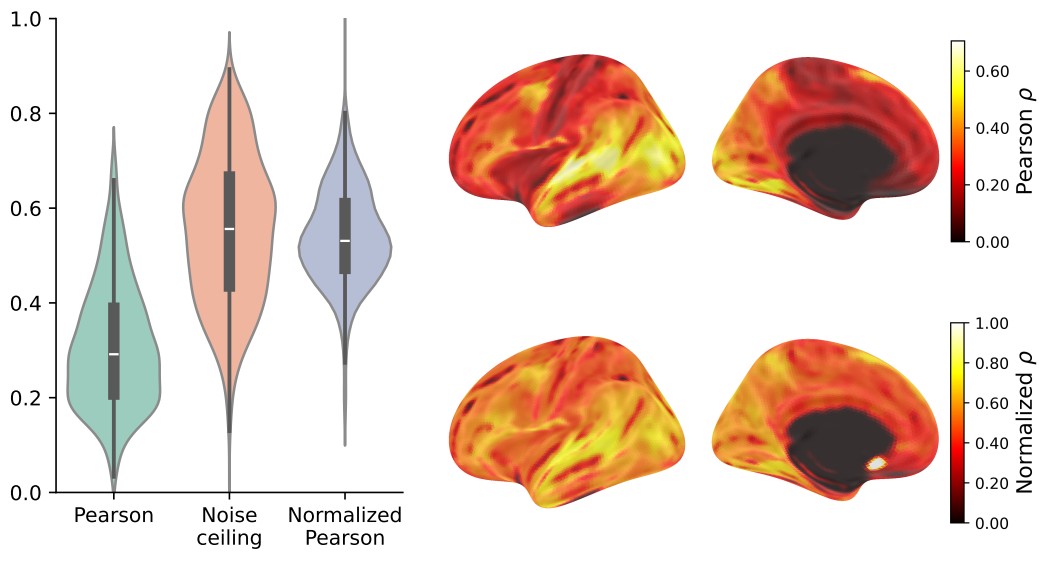

(a) Distribution across parcels          (b) Cortical projection of the encoding scores

Figure 3: **Our model predicts responses significantly across the whole brain.**
(a) We report a histogram of unnormalized and normalized encoding scores (see definition in eq. (1)) of the 1000 parcels, averaged across all validation sessions. All 1000 parcels are predicted significantly above chance ($q(\text{FDR}) < 10^{-3}$, block-wise permutation test across sessions).
(b) We project the encoding scores averaged across subjects onto the `fsaverage5` cortical surface.

### 3.3 Noise ceiling analysis

To what extent are encoding errors due to unexplainable randomness rather than model suboptimality? Following common practice in the brain encoding literature, we estimate the noise ceiling by computing the inter-trial correlation $\rho_{self}$ of the BOLD responses to the movies *Hidden Figures* and *Life*, which were viewed twice by each participant. We then define the normalized Pearson correlation of our model by dividing it by that of an ideal encoding model (Schoppe et al., 2016):

$$\rho_{norm} = \frac{\rho}{\rho_{max}}, \quad \rho_{max} = \sqrt{\frac{2}{1 + \frac{1}{\rho_{self}}}} \tag{1}$$

Our model achieves a normalized Pearson correlation of $0.54 \pm 0.1$ across all parcels (fig. 3a): in other words, it captures roughly half of explainable variance on average for the movies considered. The fairly small inter-quartile range reflects the fact that our model is rather consistent across brain areas. The highest values are achieved in the auditory and language processing cortices, where our model captures above 80% of explainable variance (fig. 3b).

### 3.4 The benefit of multimodality

To what extent do the three modalities combined by TRIBE contribute to encoding performance? We address this question in fig. 4a, by assessing the encoding performance of TRIBE retrained with various modalities ablated. When training on a single modality, TRIBE achieves substantially lower encoding scores. We observe that the text modality achieves the lowest average encoding score with 0.22, followed by audio at 0.24 and video at 0.25. When combining any two modalities together, the encoding scores rise significantly compared to the unimodal models: the best bimodal model, obtained by combining text and video, achieves an encoding score of 0.30. Finally, combining the three modalities together yields an additional boost, bringing the value to 0.31. This hints to the fact that each modality plays a complementary role.

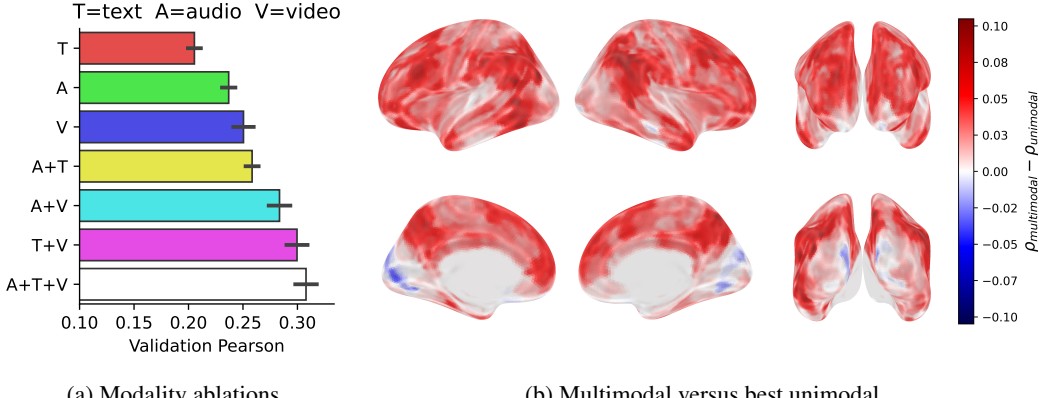

(a) Modality ablations
(b) Multimodal versus best unimodal

Figure 4: **Multimodal encoders outperform unimodal encoders.**
(a) We compare the encoding scores of encoders trained on a subset of modalities in the same conditions. Error bars denote s.e.m across subjects.
(b) For each parcel of the first subject, we compute the difference in encoding score between the multimodal encoder and the best out of the three unimodal encoders, then project the results onto the `fsaverage5` cortical surface.

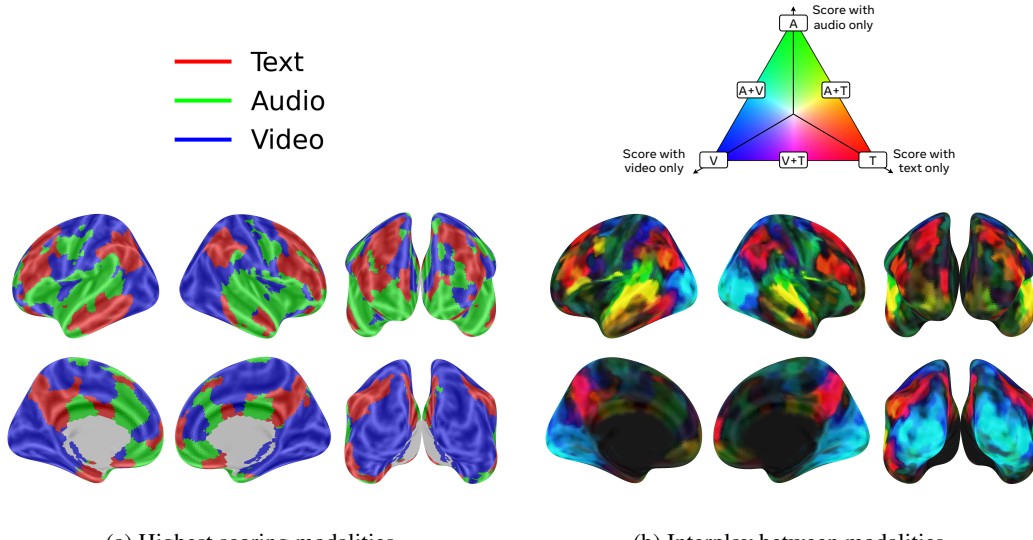

(a) Highest scoring modalities
(b) Interplay between modalities

Figure 5: **The modalities and their interplay map onto the expected brain areas.**
(a) For each parcel of the first subject, we assess the encoding contribution of each modality by masking the two other modalities, and color-code according to the highest contribution.
(b) We color-code each parcel using an RGB encoding where red, green and blue intensities are determined by the encoding score of the model with text, audio and video unmasked, respectively. For readability, we limit ourselves to bimodal interactions by substracting the smallest of the three contributions. Red, blue and green areas correspond to unimodal areas well encoded by text, audio and video respectively, while magenta, yellow and cyan correspond to bimodal areas well encoded by text-video, text-audio and video-audio interactions respectively.

In which areas does multimodality yield the strongest gains? In fig. 4b, we compare for each parcel the encoding score of the multimodal encoder with that of the best of the three unimodal encoders. We observe that the multimodal encoder consistently outperforms the unimodal models, especially in associative areas such as the prefrontal or parieto-occipito-temporal cortices (up to 30% increase in

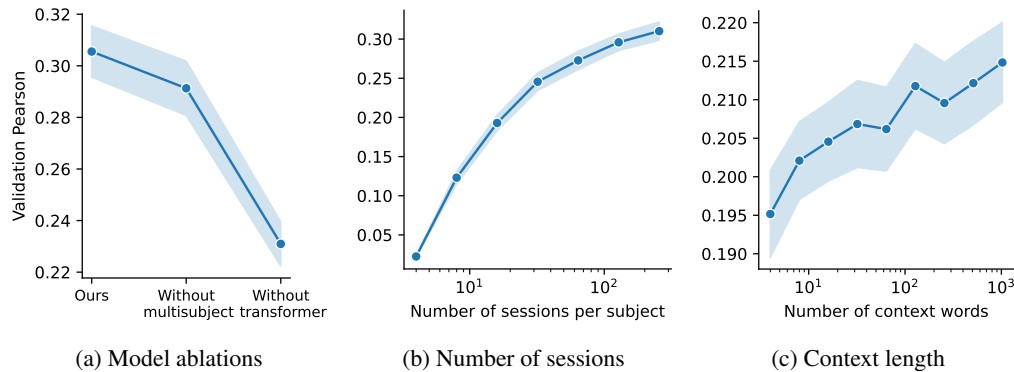

(a) Model ablations      (b) Number of sessions      (c) Context length

Figure 6: **Ablations and scaling laws of our model.**
(a) We compare the results of our model with those achieved without multi-subject training or without the transformer model.
(b) We report the results obtained by the multimodal encoder as we increase the number of sessions used in the training set.
(c) We report the results obtained by the text-only encoder as we increase the context length of the language model used to extract features (Llama-3.2-3B).
In all panels, shaded regions indicate the s.e.m over the four subjects.

encoding score). Interestingly, the multimodal model performs less well than the vision-only model in the primary visual cortex, which is highly specific to visual features.

Together, these results demonstrate that our multimodal encoder effectively captures interactions between modalities, which improves whole-brain decoding.

## 3.5 The interplay between modalities

Which brain areas is dominated by unimodal or multimodal representations? To address this question, we single out modalities by probing our multimodal model with all modalities masked off except one (using the same procedure as for the dropout described in section 2).

In fig. 5a, we color each parcel of the cortex according to the modality that achieves the highest encoding score via this procedure. The three modalities each cover broad regions: audio predominates near the temporal gyrus, video predominates in the occipital cortex and parts of the parietal cortex, while the text feature, which presumably contains the most semantic information, predominates in large parts of the parietal and prefrontal lobes.

To study the interplay between modalities, we then overlay the contribution of the three modalities using an RGB encoding where red, green and blue respectively represent the encoding scores achieved solely with text, audio and video (fig. 5b). To make hues more visible, we restrict our analyses to unimodal and bimodal interactions by substracting the smallest contribution among the three modalities. We observe interesting bimodal associations in some key areas: in particular, text+audio (yellow) in can be observed in the superior temporal lobe and video+audio (cyan) can be observed in the ventral and dorsal visual cortices.

Overall, these observations provide new insights on how multisensory integration may occur in the human cortex.

## 3.6 Ablations and scaling laws

In fig. 6a, we demonstrate the importance of using a transformer and a multi-subject training scheme: the encoding score drops from 0.31 to 0.29 when training separately for each subject, and down to 0.23 when removing the transformer. In fig. 6b, we show how the encoding score scales with the amount of recordings in the training set. We observe a strong increasing trend which has not reached a plateau, in line with recent work (Antonello et al., 2023). Finally, in fig. 6c, we show that increasing the context length used for the language model words strongly enhances encoding performance,

without any plateau even at very long context lengths of 1024 words. This confirms that TRIBE effectively captures high-level semantic features, far beyond the word and sentence level.

## 4 DISCUSSION

In this work, we make a step towards an integrative model of the brain during naturalistic perception by training an encoding model on an unprecedently-large fMRI dataset of participants watching videos. Crucially, our model is the first encoding pipeline which is simultaneously nonlinear, multisubject and multimodal: our ablations demonstrate the importance of these three aspects for encoding, especially in associative cortices. Our model achieves the first place in the Algonauts 2025 brain encoding competition, and scaling laws suggest that encoding performance increases systematically with the number of recordings, holding promise for further improvements with larger datasets.

**Limitations**     There are several remaining limitations to our work. First, our approach currently operates on a coarse parcellation of brain areas – reducing hundreds of thousands of voxels to 1,000 cortical parcels. This design choice, introduced by the Algonauts2025 challenge, likely increases the signal-to-noise ratio by smoothing out the responses spatially, and certainly reduces computational cost, which is important for the whole-brain prediction setting considered here. However, this approach limits the spatial resolution of our model, which inherently prevents it from capturing highly localized phenomena. Adapting our model for voxel-level prediction is an important avenue for future work. Second, our current approach is limited to fMRI data. Consequently, the precise temporal dynamics of neuronal activity, and the exact neural assemblies underlying these macroscopic signals, remain, here, unresolved. Applying a similar approach for electro- or magneto-encephalographic signals could prove invaluable in this regard. Third, while the volume of recording per participant in the study considered here is particularly large, only four participants were included: extending and improving our results on a larger pool of participants is an important next step. In particular, whether TRIBE can generalize to unseen subjects, either in a zero-shot or few-shot setting, is an important question for future work, which will likely require naturalistic datasets with a larger pool of participants, such as that of the Human Connectome Project (Van Essen & Ugurbil, 2012). Fourth, our model predicts responses deterministically from the perceptual inputs, at odds with the brain – for example: in the absence of stimuli, the default mode network is expected to display a complex dynamical pattern (Raichle, 2015), which cannot be captured by our model. Transitioning towards generative methods such as diffusion models would be necessary to capture such phenomena. Finally, the present approach remains limited to perception and comprehension. Behavior, memory and decisions are other important cognitive components to integrate into the present model.

**Broader impact**     Building a model able to accurately predict human brain responses to complex and naturalistic conditions is an important endeavour for neuroscience. Not only does this approach open the possibility of exploring cognitive abilities (e.g. theory of mind, humour) that are challenging to isolate with minimalist designs, but they will eventually be necessary to evaluate increasingly complex models of cognition and intelligence. In addition, our approach forges a path to (1) integrate the different sub-fields of neuroscience into a single framework and to (2) develop in silico experimentation (Jain et al., 2024), where in vivo experiments could be complemented and guided by the predictions of a brain encoder. While the exploration of this paradigm falls beyond the scope of this technical report, we believe that epistemology is ordered: interpretation and scientific insights are most legitimate if they derive from the model that makes the best prediction. In this regard, the first place achieved by TRIBE in the Algonauts 2025 competition gives scientific credit to our approach.

**Ethics statement** We aim for our whole-brain encoding framework to chart a viable path toward a foundational model of cognition – one that bridges the diverse avenues of neuroscientific research. We do however call for caution concerning the negative avenues of this line of research: brain response prediction could become a lucrative endeavour which may exploit our cognitive weaknesses, e.g. to amplify harmful consumption habits (Murphy et al., 2008).

**Reproducibility statement** We provide a fully runnable codebase to reproduce our results at the following address: `https://anonymous.4open.science/r/algonauts-2025-C63E`.

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

| Hyperparameter | Base value | Other values used for ensembling |
|---|---|---|
| Number of epochs | 15 | |
| Number of TRs per window | 100 | |
| Window jitter | 10s | |
| Learning Rate | $10^{-4}$ | |
| Batch Size | 16 | |
| Optimizer | AdamW | |
| Scheduler type | OneCycleLR (cosine) | |
| Scheduler warmup phase | 10% | |
| Stochastic weight average epochs | 8 | |
| Dropout | 0 | |
| Weight Decay | 0 | |
| Hidden Size | 3072 | |
| Text model | Llama-3.2-3B | |
| Audio model | Wav2Vec-Bert-2.0 | |
| Video model | V-JEPA-2-Gigantic-256 | |
| Loss | MSE | Pearson, SmoothL1, HuberLoss |
| Modality Dropout | 0.2 | 0.0, 0.4 |
| Layer groups | [0.5, 0.75, 1] | [0,0.5,1], [0.5, 1], [0, 0.2, 0.4, 0.6, 0.8, 1.] |
| Layer extraction | group by intervals | extract single layers |
| Layer aggregation | concatenate | average |
| Modality aggregation | concatenate | average |
| Use subject embedding | ✓ | ✗ |

Table 3: Hyperparameters used in our model

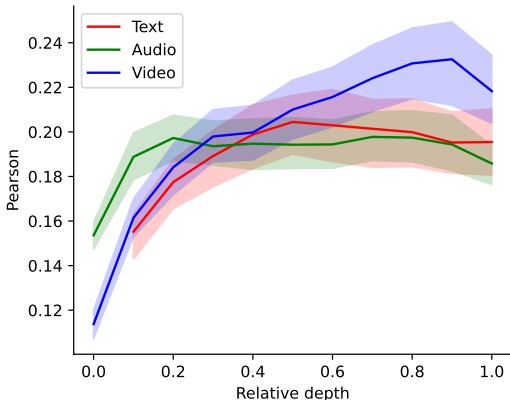

Figure 7: **Layerwise analysis of encoding scores.** For each modality, we train unimodal encoders on top of the intermediate representations of the corresponding foundational model at various relative depths (0 standing for the first layer and 1 standing for the last layer of the model).

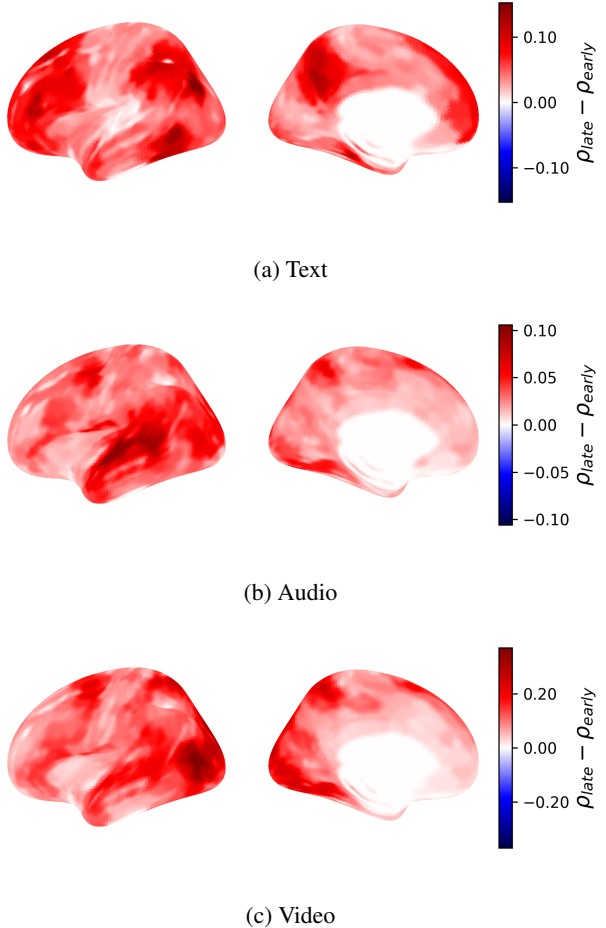

(a) Text

(b) Audio

(c) Video

Figure 8: **Late layers better encode high-level cortices than early layers.** For each modality, we report the difference between the encoding score of the last layer and the first layer of the corresponding foundational model.

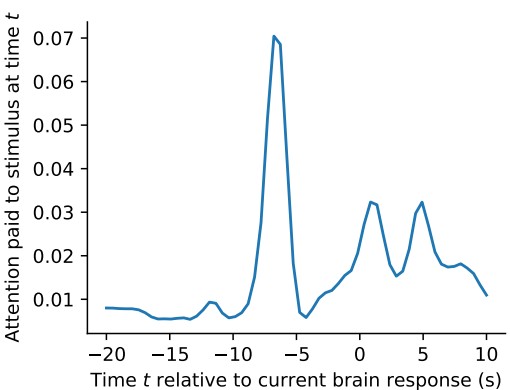

Figure 9: **Attention maps are reminiscent of the hemodynamical response function.** For various times $t$ spanning the stimulus window, we report the attention weight on the surrounding timesteps. The results are averaged over all $t$ in the window, and across all windows of the validation set.

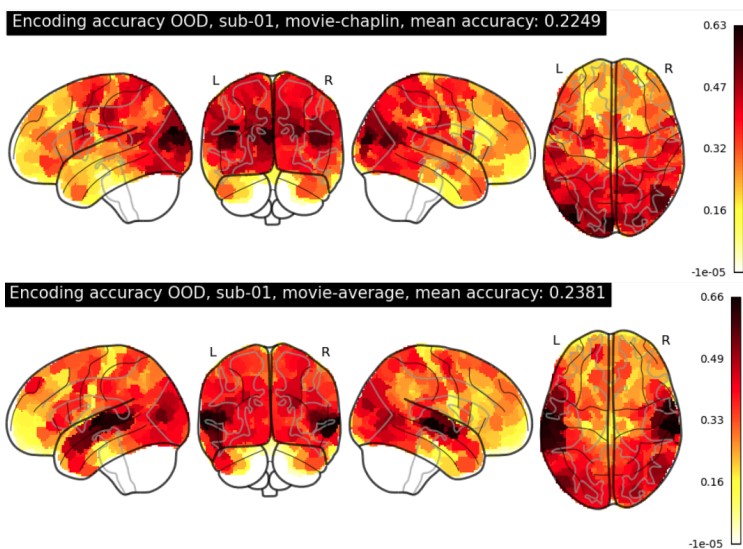

Figure 10: **Silent movies are better predicted in the visual cortex and less well predicted in the auditory cortex.** Top: results for the first subject on the Charlie Chaplin movie. Bottom: averaged across all OOD movies. Since the silent Chaplin movie belongs to the private test set of the competition, we do not have access to the ground truth labels and report screenshots of results provided by the competition host for the first subject.

LICENSES

HuggingFace models:

- Video-JEPA 2: Apache (`https://github.com/facebookresearch/vjepa2/blob/main/LICENSE`)
- Wav2Vec-Bert-2.0: MIT (`https://huggingface.co/datasets/choosealicense/licenses/blob/main/markdown/mit.md`)
- LLama-3.2-3B: llama3.2 (`https://huggingface.co/meta-llama/Llama-3.2-1B/blob/main/LICENSE.txt`)

Packages:

- `x-transformers`: MIT (`https://github.com/lucidrains/x-transformers/blob/main/LICENSE`)
- `nilearn`: BSD (`https://github.com/nilearn/nilearn/blob/main/LICENSE`)
- `pytorch`: `https://github.com/pytorch/pytorch/blob/main/LICENSE`

Datasets:

- Courtois NeuroMod: CC0 (`https://creativecommons.org/publicdomain/zero/1.0/legalcode`)

