# OpenReview forum: "TRIBE: TRImodal Brain Encoder for whole-brain fMRI response prediction"
_ICLR.cc/2026/Conference — ICLR 2026 Poster_

### Official Review · Reviewer_wQpp · 2025-10-27

**Soundness:** 3
**Presentation:** 3
**Contribution:** 3
**Rating:** 8
**Confidence:** 4

**Summary:**

TRIBE presents and develops a neural network to predict whole-brain fMRI responses to naturalistic videos through integrated text, audio, and visual processing. The work addresses three critical limitations of prior encoding models: their reliance on linear mappings between AI and brain representations, subject-specific training that cannot leverage cross-individual similarities, and restriction to single modalities. The architecture extracts features from pretrained foundation models (Llama-3.2-3B for text, Wav2Vec-BERT-2.0 for audio, V-JEPA-2 for video), synchronizes them at 2 Hz, compresses layerwise representations, projects to a shared 1024-dimensional space, and processes concatenated multimodal embeddings through an 8-layer transformer with learnable subject and positional embeddings before mapping to 1000 cortical parcels.

Training on over 80 hours of fMRI per subject from the Courtois NeuroMod dataset, TRIBE achieves first place among 263 teams in the Algonauts 2025 competition, achieving a substantial margin over competitors. All 1000 parcels show significant prediction accuracy, and the model captures 54% of explainable variance on average, exceeding 80% in auditory and language cortices. Performance remains robust on highly out-of-distribution stimuli including animations, nature documentaries, and silent films. Ablations demonstrate that multimodality provides greatest benefits in associative cortices (up to 30% improvement over best unimodal baseline) while vision-only features slightly outperform in primary visual cortex. Both the transformer architecture and multi-subject training prove essential, and scaling analyses show continued improvement with more data and longer linguistic context without saturation.

the spatial averaging required for computational tractability likely discards retinotopic information critical for low-level visual encoding. Nevertheless, the work represents significant methodological progress by demonstrating that nonlinear integration of foundation model representations within a unified subject-general architecture can effectively predict whole-brain responses to complex naturalistic stimuli, providing a foundation for more comprehensive models of human cognition.

**Strengths:**

1) The model’s architecture is not a brute-force fusion but a theoretically motivated hierarchy: frozen modality experts (V-JEPA2, Wav2Vec2-BERT, LLaMA3.2) adapted into a shared latent space, followed by a transformer capturing temporal and intersubject alignment. The multisubject encoder and modality dropout mechanisms seem particularly well thought out, improving both biological plausibility and statistical efficiency

2) The multimodal integration analysis is both methodologically clear and scientifically meaningful: ablations show monotonic gains from unimodal to bimodal to trimodal configurations (0.22 → 0.31 Pearson), empirically supporting the hypothesis that integrated sensory embeddings yield richer cortical alignment.

3) TRIBE achieves a normalized Pearson correlation of ~0.54 ± 0.1 across 1,000 parcels, explaining more than half the variance in fMRI responses. The consistency across subjects and cortical regions is noteworthy,

**Weaknesses:**

1) V-JEPA activations are averaged over patches to manage compute, and the authors themselves expect degradation in low-level, retinotopic cortex. That choice complicates interpreting modality-dominance maps in early vision. It would make the paper stronger if the authors can provide some clarity here.

2) The normalized Pearson metric (Eq. 1, claiming TRIBE captures "54% of explainable variance") relies on test-retest reliability (ρself) computed from only two repeated movies: Hidden Figures and Life. While I acknowledge that this follows standard practice in the field, Table 2 reveals substantial performance heterogeneity across genres: raw Pearson correlations span nearly 2× from Friends (0.32) to Charlie Chaplin (0.17). Given that different stimulus classes likely have different signal-to-noise characteristics (e.g., silent films vs. dialogue-heavy sitcoms, etc.), the reliability ceiling may also vary systematically across genres. Without demonstrating that Hidden Figures and Life yield consistent ρself estimates, or that their ceiling generalizes to the broader stimulus distribution, the "54%" figure may not accurately represent explainable variance for all content types in the test set. This perhaps limits the interpretation of normalized scores, particularly for out-of-distribution evaluation. A simple robustness check comparing ρself between the two repeated movies, or per-parcel ceiling variability, would  help to strengthen confidence in this metric.

3)  The pipeline downsamples all modalities to 2 Hz, processes via transformer, and pools to TR (1.49s), but nowhere mentions hemodynamic response function (HRF) convolution. The HRF introduces ~5-6 second lag and temporal smoothing between neural activity and BOLD. While the transformer's large receptive field may implicitly learn such dynamics, three concerns arise: (1) downsampling audio from 50 Hz → 2 Hz may discard rapid transients (phonemes, prosody) that could improve prediction when properly lagged/smoothed to TR resolution, (2) interpretability suffers i.e. is the model learning neural dynamics requiring HRF or fitting BOLD directly?, and (3) no ablation (Table 3) tests whether explicit HRF convolution would improve performance. Given that competitors likely vary in temporal modeling choices, this could represent either a hidden performance limiter or an architectural shortcut that the transformer adequately compensates for but the paper provides no evidence to distinguish these possibilities.

**Questions:**

1) Section 2.3 describes layer normalization after projecting each modality to 1024-D. For replicability, please clarify: (1) Are raw embeddings from each foundation model normalized (e.g., z-scored per modality over time) before the linear projection, or do they enter the projection layer unnormalized? Given different embedding dimensions and scales across models, this choice affects how the network learns to balance modalities.

2) The paper tests multiple layer-grouping strategies (Table 3) but does not report which configurations perform best or where. Even a summary-level analysis would be valuable: Do models extracting early vs. late layers from each backbone receive differential ensemble weights in sensory vs. associative regions? I recognize that the 1000-model ensemble with confounded hyperparameters complicates attribution, but even coarse patterns (e.g., "models grouping early V-JEPA layers achieved higher weights in V1") would provide interpretability consistent with standard brain encoding practice. If full layer-wise decomposition is infeasible, reporting which layer-grouping strategies from Table 3 achieved highest overall performance can help inform architectural choices for future works.

3) Table 2 shows substantial performance variation across stimulus types (0.32 for Friends vs. 0.17 for Chaplin). While Figure 4a provides modality ablations averaged across validation data, exploring whether multimodal benefits vary by stimulus characteristics could offer additional insight, e.g., whether audio contributes less for silent films. I recognize this may require re-running unimodal models on each test movie and that the limited sample (n=1 per genre) constrains statistical conclusions, but even illustrative case comparisons could be informative.

---

> ### Author Response · Authors · 2025-11-21
> **Rebuttal [part 1]**
>
> We thank the reviewer for their extremely careful review and thoughtful suggestions, which have led us to include three additional figures which we believe significantly strengthen our paper.
>
> > V-JEPA activations are averaged over patches to manage compute, and the authors themselves expect degradation in low-level, retinotopic cortex. That choice complicates interpreting modality-dominance maps in early vision. It would make the paper stronger if the authors can provide some clarity here.
>
> This is indeed a core limitation of our model, which is built as a whole-brain predictor rather than a specialized model for vision. Note that as shown in figure 4, in early vision areas, our model is actually outperformed by unimodal encoders. We believe that designing efficient ways to compress the spatial information in images is an important avenue to improve modelling in such areas, but is beyond the scope of our work.
>
> > The normalized Pearson metric (Eq. 1, claiming TRIBE captures "54% of explainable variance") relies on test-retest reliability (ρself) computed from only two repeated movies: Hidden Figures and Life. While I acknowledge that this follows standard practice in the field, Table 2 reveals substantial performance heterogeneity across genres: raw Pearson correlations span nearly 2× from Friends (0.32) to Charlie Chaplin (0.17). Given that different stimulus classes likely have different signal-to-noise characteristics (e.g., silent films vs. dialogue-heavy sitcoms, etc.), the reliability ceiling may also vary systematically across genres. Without demonstrating that Hidden Figures and Life yield consistent ρself estimates, or that their ceiling generalizes to the broader stimulus distribution, the "54%" figure may not accurately represent explainable variance for all content types in the test set. This perhaps limits the interpretation of normalized scores, particularly for out-of-distribution evaluation. A simple robustness check comparing ρself between the two repeated movies, or per-parcel ceiling variability, would help to strengthen confidence in this metric.
>
> We thank the reviewer for this comment. We ran the same analysis separately for the two movies and did not find significant differences. However, we agree that the figure 54% must be taken with a grain of salt: what we wanted to convey is an order of magnitude, without assuming that this generalizes to other distributions. We now clarify this by removing the explicit figure 54% from the conclusion: *“in other words, it captures roughly half of explainable variance on average for the movies considered here”.*
>
> > The pipeline downsamples all modalities to 2 Hz, processes via transformer, and pools to TR (1.49s), but nowhere mentions hemodynamic response function (HRF) convolution. The HRF introduces ~5-6 second lag and temporal smoothing between neural activity and BOLD. While the transformer's large receptive field may implicitly learn such dynamics, three concerns arise: (1) downsampling audio from 50 Hz → 2 Hz may discard rapid transients (phonemes, prosody) that could improve prediction when properly lagged/smoothed to TR resolution
>
> We thank the reviewer for bringing up this interesting discussion. Interestingly, the other top competitors actually follow the design introduced by the competition: the stimuli features are sampled at the repetition time of the scanner, i.e. at a frequency of 0.66 Hz, even lower than our 2Hz (this may in fact explain part of the success of TRIBE compared to others). We agree with the reviewer that increasing this sampling frequency does gather more information from the stimuli, but also leads to a larger memory footprint. We now include a discussion of this tradeoff in the methods: *“Since the number of input tokens is $N\times TR \times f$, at fixed memory budget, there is a tradeoff between increasing the stimulus sampling frequency $f$ and the window duration $N$. Through a grid search with frequencies ranging from $0.5$ to $10$\,Hz at fixed number of input tokens, we found $f=2$\,Hz and $N=100$ to give the best results.”*

---

> > ### Author Response · Authors · 2025-11-21
> > **Rebuttal [part 2]**
> >
> > > (2) interpretability suffers i.e. is the model learning neural dynamics requiring HRF or fitting BOLD directly?, and (3) no ablation (Table 3) tests whether explicit HRF convolution would improve performance. Given that competitors likely vary in temporal modeling choices, this could represent either a hidden performance limiter or an architectural shortcut that the transformer adequately compensates for but the paper provides no evidence to distinguish these possibilities.
> >
> > We did not experiment convolving the stimulus embeddings with an HRF, under the rationale that HRFs are expected to vary across subjects and conditions: hence, we believe it is better to let the transformer learn the appropriate smoothing function itself. Motivated by this comment, we ran an attention analysis to clarify what the resulting “learnt” HRF looks like, reported in figure 9: for each point t in the window, we plot the attention weights A_tt’ for t’ ranging between -20 seconds and +10 seconds, and average the result over t and across batches in the validation set. The resulting attention curve peaks between -10 and -4 seconds, consistently with what could be expected from the HRF. Interestingly, we also observe two secondary peaks around t=0, which could be due to the encoding of future predictions, but prefer not to speculate on this as attention maps of Transformers without registers are known to also exhibit artefacts [1].
> > We now report in the main text: *“Since the transformer has access to all timesteps, we do not follow the common practice in linear encoding of convolving the inputs by a temporal response function. Instead, the attention mechanism selects the most relevant timesteps: an analysis presented in fig.9 shows that the attention weights peak between 5 and 10 seconds relative to the current timestep, which is consistent with the expected hemodynamic response function.”*
> >
> > [1] Darcet, Timothée, et al. "Vision transformers need registers." arXiv preprint arXiv:2309.16588 (2023).
> >
> > >Section 2.3 describes layer normalization after projecting each modality to 1024-D. For replicability, please clarify: (1) Are raw embeddings from each foundation model normalized (e.g., z-scored per modality over time) before the linear projection, or do they enter the projection layer unnormalized? Given different embedding dimensions and scales across models, this choice affects how the network learns to balance modalities.
> >
> > The raw embeddings are not normalized - the normalization is precisely left to the layer normalization module (that is, before the concatenation), as we cannot be sure whether each modality is of same importance.

---

> > > ### Author Response · Authors · 2025-11-21
> > > **Rebuttal [part 3]**
> > >
> > > > The paper tests multiple layer-grouping strategies (Table 3) but does not report which configurations perform best or where. Even a summary-level analysis would be valuable: Do models extracting early vs. late layers from each backbone receive differential ensemble weights in sensory vs. associative regions? I recognize that the 1000-model ensemble with confounded hyperparameters complicates attribution, but even coarse patterns (e.g., "models grouping early V-JEPA layers achieved higher weights in V1") would provide interpretability consistent with standard brain encoding practice. If full layer-wise decomposition is infeasible, reporting which layer-grouping strategies from Table 3 achieved highest overall performance can help inform architectural choices for future works.
> > >
> > > We thank the reviewer for this useful suggestion. Motivated by this, we ran a layerwise analysis for each modality, which we now report in figures 7 and 8. We recover the expected result that the pearson score peaks in the second half of the relative depth for each modality, and late layers are particularly strong in high-level cortices (note however that they are not worse in low-level cortices, which is why our ensembling configurations always include later layers). We now add in the methods section: *“As shown in fig. 7, for each modality, the embeddings coming from deeper layers of the corresponding foundational model yield better encoding performance, especially in associative cortices (fig. 8),, and the best configuration was found to be $L=2$ groups ranging from relative depths 0.5 to 0.75, and 0.75 to 1.”*
> > >
> > > > Table 2 shows substantial performance variation across stimulus types (0.32 for Friends vs. 0.17 for Chaplin). While Figure 4a provides modality ablations averaged across validation data, exploring whether multimodal benefits vary by stimulus characteristics could offer additional insight, e.g., whether audio contributes less for silent films. I recognize this may require re-running unimodal models on each test movie and that the limited sample (n=1 per genre) constrains statistical conclusions, but even illustrative case comparisons could be informative.
> > >
> > > Unfortunately, we do not have access to the test data for these stimuli as it corresponds to the final test set of the competition which is now closed. However, the competition does provide encoding score maps, which clearly show that for the silent movie, language processing cortices are significantly less well encoded than on average across movies. We now report this in figure 10 and cite this figure as follows in the main text: *“silent black-and-white movies (0.1686 for \textit{Charlie Chaplin}), where language processing cortices are very poorly encoded (see fig.10).”.*

---

### Official Review · Reviewer_dPCk · 2025-10-29

**Soundness:** 3
**Presentation:** 3
**Contribution:** 3
**Rating:** 8
**Confidence:** 3

**Summary:**

This paper introduces TRIBE, a deep learning model that non-linearly integrates text, audio, and video features using a transformer to predict whole-brain fMRI responses to naturalistic videos. Its primary strength is its state-of-the-art performance, demonstrated by winning the Algonauts 2025 competition. The work is significant as it simultaneously addresses some key limitations of previous encoding models.

**Strengths:**

1. The motivation of this article is very good, as it analyzes the entire brain information from a multimodal perspective. It is of great practical significance and better aligns with the data processing procedures in the era of large models. Therefore, it is also very beneficial for researching more general brain foundation data.

2. The experiments are well-conducted, and the performance is also good. Its 1st-place ranking out of 263 teams in a competitive benchmark is the strongest and most objective evidence for its effectiveness.

3. Besides the model, this paper provides valuable neuroscientific insights, showing that multimodality provides the largest gains in associative cortices and reveals expected unimodal/bimodal dominance in different brain regions.

**Weaknesses:**

1. The article does not disclose or discuss the complexity of the model. To the best of my knowledge, many previous brain decoding projects employed relatively small models. However, the current method employs multiple pre-trained large models, and it should provide the overall size of the model so that others can evaluate and use it.

2. Starting from line 48, the first two motivations actually involve many models that no longer use simple regression. Moreover, numerous recent studies have focused on multi-subject scenarios (e.g.,[1][2][3][4]), and it is essential to accurately describe the current research status .

(In fact, the main weakness has been honestly stated by the authors in the "limitations" section.)

[1] Mindeye2: Shared-subject models enable fmri-to-image with 1 hour of data

[2] Umbrae: Unified multimodal brain decoding

[3] CLIP-MUSED: CLIP-Guided Multi-Subject Visual Neural Information Semantic Decoding

[4] Toward Generalizing Visual Brain Decoding to Unseen Subjects

**Questions:**

1. The article mentions in the limitations section that the number of participants is currently small. Another concern I have is whether the model can be generalized to new subjects as discussed in [1]. If the brain decoding model is used in practice, it is unrealistic for each user to collect a large amount of their own data. With more modalities and increased data volume, can the model demonstrate its generalization ability across different subjects? (For example, a model trained on subjects 1-3 should also work for subject 4.)

2. I also have doubts about the accuracy because this task involves predicting brain activity in response to different stimuli. However, even when the same subjects receive identical stimuli, their brain activity often varies, especially with temporal stimuli like audio and video. Since I lack the relevant background knowledge in neuroscience, I'm unsure if this mapping is reasonable. Nevertheless, this issue is more related to the competition's design. The authors can provide an explanation based on the specific circumstances.

[1] Toward Generalizing Visual Brain Decoding to Unseen Subjects

---

> ### Author Response · Authors · 2025-11-21
> **Rebuttal**
>
> We thank the reviewer for their thoughtful comments and suggestions. Please find our responses below.
>
> > The article does not disclose or discuss the complexity of the model. To the best of my knowledge, many previous brain decoding projects employed relatively small models. However, the current method employs multiple pre-trained large models, and it should provide the overall size of the model so that others can evaluate and use it.
>
> We thank the reviewer for pointing this out: indeed, we forgot to include model size in our discussion on compute requirements, and now report the following:
> *“These models respectively contain 3B, 600M and 700M parameters, and feature extraction is completed in 24 hours on 128 V100 GPUs with 32GB of VRAM. The TRIBE model itself contains contains 980M trainable parameters, and the full training loop lasts 24 hours on a single such GPU.”*
>
> > Starting from line 48, the first two motivations actually involve many models that no longer use simple regression. Moreover, numerous recent studies have focused on multi-subject scenarios (e.g.,[1][2][3][4]), and it is essential to accurately describe the current research status.
>
> We would like to point out that these references are decoding models rather than encoding models. We do in fact emphasize at the beginning of the related works section that such scenarios have been explored in the decoding literature (and in fact cite reference 2): *“While there has been recent research on deep learning for multimodal brain decoding (Dahan et al., 2025; Xia et al., 2024), there currently exists no equivalent for brain encoding”*. We now also report the 3 other references provided by the reviewer to this sentence, as they are useful to highlight the gap between the two lines of research.
>
> > The article mentions in the limitations section that the number of participants is currently small. Another concern I have is whether the model can be generalized to new subjects as discussed in [1]. If the brain decoding model is used in practice, it is unrealistic for each user to collect a large amount of their own data. With more modalities and increased data volume, can the model demonstrate its generalization ability across different subjects? (For example, a model trained on subjects 1-3 should also work for subject 4.)
>
> We thank the reviewer for bringing up this valid concern. As it happens, we have been exploring cross-subject generalization in follow-up work, following two paradigms: (i) zero-shot generalization, where we average the subject-conditional layer to predict unseen subjects, and (ii) finetuning, where we fine-tune the subject layer on a small amount of data for new subjects. However, we believe that this research falls outside of the scope of the current paper: for now, we add in the following discussion in the limitations section: *“In particular, whether TRIBE can generalize to unseen subjects, either in a zero-shot or few-shot setting, is an important question for future work, which will likely require naturalistic datasets with a larger pool of participants, such as that of the Human Connectome Project~\cite{van2012future}.”.*
>
> >I also have doubts about the accuracy because this task involves predicting brain activity in response to different stimuli. However, even when the same subjects receive identical stimuli, their brain activity often varies, especially with temporal stimuli like audio and video. Since I lack the relevant background knowledge in neuroscience, I'm unsure if this mapping is reasonable. Nevertheless, this issue is more related to the competition's design. The authors can provide an explanation based on the specific circumstances.
>
> This is a very interesting remark: indeed, the regression targets are not deterministically related to the inputs. This is particularly striking if we consider the setting where no stimuli are presented to the participant: our model would output a constant prediction, whereas it is known from neuroscience that the brain undergoes dynamics in the default mode network. We now add the following discussion in the limitations section:
> *“Our model predicts responses deterministically from the perceptual inputs, at odds with the brain -- for example: in the absence of stimuli, the default mode network is expected to display a complex dynamical pattern~\cite{raichle2015brain}, which cannot be captured by our model. Transitioning towards generative methods such as diffusion models would be necessary to capture such phenomena.”*

---

### Official Review · Reviewer_DyxP · 2025-11-05

**Soundness:** 3
**Presentation:** 2
**Contribution:** 3
**Rating:** 6
**Confidence:** 2

**Summary:**

The paper presents TRIBE, a model that predicts brain activity while people watch movies, using features from large pretrained models for text, audio, and video. By combining these three types of information and processing them with a transformer, the model can capture how different parts of the brain respond to complex, real-world stimuli. TRIBE achieves first place in a public competition, showing strong results across the whole brain, especially in regions that integrate multiple senses. The work suggests that using multimodal data and deep learning can help build more general models of brain function.

**Strengths:**

1. Combines text, audio, and video features for brain prediction, which covers more brain regions than single-modality models.
2. Achieves clear state-of-the-art results on a public benchmark, outperforming all other competitors by a noticeable margin.
3. The model design is practical and scalable, using existing large models and a transformer to handle real-world, naturalistic data.

**Weaknesses:**

1. The model relies on combining a large number of models (ensemble), so it’s unclear how well a single model works in practice.
2. Some details in the paper are inconsistent, for example the feature dimensions in Figure 2 don’t match the numbers in the methods section, and the number of teams is sometimes 262 and sometimes 263.

**Questions:**

See weakness.

---

> ### Author Response · Authors · 2025-11-21
> **Rebuttal**
>
> We thank the reviewer for these valuable remarks, and are glad they appreciated our work. We now include three additional figures as suggested by other reviewers, which the reviewer may be interested in reading about.
>
> > The model relies on combining a large number of models (ensemble), so it’s unclear how well a single model works in practice.
>
> Aside from the official competition results presented in the tables, all results presented in the paper concern a single model. While the gap between the encoding score of the single model (0.3091)  and the ensembled model (0.3195) is significant, the single model still performs well in practice, and in particular encodes all 1000 parcels significantly better than chance as demonstrated in figure 3.
>
> > Some details in the paper are inconsistent, for example the feature dimensions in Figure 2 don’t match the numbers in the methods section, and the number of teams is sometimes 262 and sometimes 263.
>
> We thank the reviewer for pointing out these mistakes, which have been corrected in the updated manuscript.

---

> > ### Comment · Reviewer_DyxP · 2025-11-26
> >
> > Thanks for the authors' response; my review remains the same.

---

### Public Comment · ~B.Teja_sree1 · 2025-11-20
**Anonymity violation in submission**

Dear Area Chair,

I would like to report an anonymity violation in this submission. The paper explicitly states that the authors were the **first-place winners of the Algonauts Challenge**. This information can be directly used to identify the author team, since their challenge solution and identities were publicly presented at CCN-2026.

This also risks introducing bias for reviewers, as it makes clear who the authors are in the competition. Furthermore, it is not necessary for the authors to present this as a “challenge-winning” entry in this context; since this is a conference and more research-driven, they could simply state that they were among the top performers without revealing their exact ranking or identity.

This makes the submission non-anonymous and violates the double-blind review policy.

---

> ### Author Response · Authors · 2025-11-21
> **This follows the 2026 Author Guide**
>
> Directly citing the ICLR 2026 Author Guide (https://iclr.cc/Conferences/2026/AuthorGuide):
>
> > Q: Can you explain how to treat de-anonymization in the case where a submitted paper refers to a challenge they won which can identify the authors?
>
> > It is ok to report the results on the leaderboard of a challenge. The authors can include the ranking and the name of the challenge. The reviewers will be advised to not intentionally search the authors by examining the leaderboard."
>
> > Q. I have a nearly identical version on arxiv. Does this violate the anonymity policy?
>
> > No, so long as you do not refer to it explicitly.

---

### Meta-Review · Area_Chair_diDz · 2026-01-12

**Summary:**

TRIBE presents a transformer-based architecture that integrates text, audio, and video features to predict whole-brain fMRI responses.

**Reviewer Concerns:**

Reviewers questioned the lack of an explicit Hemodynamic Response Function (HRF). The authors addressed this by providing an attention analysis suggesting that the transformer might implicitly learn the 4–10 second biological lag.
To improve transparency, the authors disclosed the model's scale (980M trainable parameters) and clarified that Layer Normalization is used to balance the diverse multimodal embeddings before integration.
Concerns regarding the "54% explainable variance" claim were met with a more nuanced discussion. The authors indicated consistent reliability across different movies but updated the manuscript to frame this figure as an average specific to the study's distribution.
Reviewers noted a performance dip in early visual areas due to spatial averaging. The authors claimed this as a deliberate trade-off. The authors' argument seems debatable.
The work provides an algorithm to fit fMRI signals and makes a contribution to this field.

**Reviewer Scores:**

It is hard to tell. The authors did provide responses that would have benefited from discussion.

---

### Decision · Program_Chairs · 2026-01-26

Accept (Poster)